# Mathematical Modelling of Cervical Precancerous Lesion Grade Risk Scores: Linear Regression Analysis of Cellular Protein Biomarkers and Human Papillomavirus *E6*/*E7* RNA Staining Patterns

**DOI:** 10.3390/diagnostics13061084

**Published:** 2023-03-13

**Authors:** Sureewan Bumrungthai, Tipaya Ekalaksananan, Pilaiwan Kleebkaow, Khajohnsilp Pongsawatkul, Pisit Phatnithikul, Jirad Jaikan, Puntanee Raumsuk, Sureewan Duangjit, Datchani Chuenchai, Chamsai Pientong

**Affiliations:** 1Division of Biopharmacy, Faculty of Pharmaceutical Sciences, Ubon Ratchathani University, Ubon Ratchathani 34190, Thailand; sureewan.b@windowslive.com; 2Division of Microbiology and Parasitology, School of Medical Sciences, University of Phayao, Phayao 56000, Thailand; 3HPV & EBV and Carcinogenesis Research Group, Khon Kaen University, Khon Kaen 40002, Thailand; 4Department of Microbiology, Faculty of Medicine, Khon Kaen University, Khon Kaen 40002, Thailand; 5Department of Obstetrics and Gynecology, Faculty of Medicine, Khon Kaen University, Khon Kaen 40002, Thailand; 6Department of Social Medicine, Phayao Hospital, Phayao 56000, Thailand; 7Department of Cytopathology, Phayao Hospital, Phayao 56000, Thailand; 8Division of Pharmaceutical Chemistry and Technology, Faculty of Pharmaceutical Sciences, Ubon Ratchathani University, Ubon Ratchathani 34190, Thailand

**Keywords:** machine learning, mathematical model, cervical cancer, biomarker, risk score, HPV

## Abstract

The current practice of determining histologic grade with a single molecular biomarker can facilitate differential diagnosis but cannot predict the risk of lesion progression. Cancer is caused by complex mechanisms, and no single biomarker can both make accurate diagnoses and predict progression risk. Modelling using multiple biomarkers can be used to derive scores for risk prediction. Mathematical models (MMs) may be capable of making predictions from biomarker data. Therefore, this study aimed to develop MM–based scores for predicting the risk of precancerous cervical lesion progression and identifying precancerous lesions in patients in northern Thailand by evaluating the expression of multiple biomarkers. The MMs (Models 1–5) were developed in the test sample set based on patient age range (five categories) and biomarker levels (cortactin, p16^INK4A^, and Ki–67 by immunohistochemistry [IHC], and HPV *E6*/*E7* ribonucleic acid (RNA) by in situ hybridization [ISH]). The risk scores for the prediction of cervical lesion progression (“risk biomolecules”) ranged from 2.56–2.60 in the normal and low–grade squamous intraepithelial lesion (LSIL) cases and from 3.54–3.62 in cases where precancerous lesions were predicted to progress. In Model 4, 23/86 (26.7%) normal and LSIL cases had biomolecule levels that suggested a risk of progression, while 5/86 (5.8%) cases were identified as precancerous lesions. Additionally, histologic grading with a single molecular biomarker did not identify 23 cases with risk, preventing close patient monitoring. These results suggest that biomarker level–based risk scores are useful for predicting the risk of cervical lesion progression and identifying precancerous lesion development. This multiple biomarker–based strategy may ultimately have utility for predicting cancer progression in other contexts.

## 1. Introduction

Almost all cervical cancers and their precursor lesions, including squamous intraepithelial lesions (SIL) and cervical intraepithelial neoplasia (CIN), are caused by persistent infection with high–risk (HR) human papillomavirus (HPV) genotypes [1,2,3,4,5]. Cytopathology reveals metaplastic cervical epithelium in approximately 20–30% of HPV–infected patients, but these features generally resolve spontaneously within one year after HPV infections [6]. In the history model of HPV–driven cervical carcinogenesis, abnormal cells gradually grow, progress to precursor lesions (i.e., CIN 3, approximating carcinoma in situ), and invade [6,7]. It is estimated that over 70% of women worldwide will be infected with HPV during their lifetime [8]. However, infections persist in fewer than 10% of women, who then experience an increased risk of developing carcinomas in situ [1,2,3]. These carcinomas gradually grow into large precancerous lesions that have a 30–50% risk of invasion over the remainder of a woman’s life [9]. Among HR–HPV genotypes, HPV16 and HPV18 confer the highest risk of carcinoma in situ and of invasive cancer [10]. Various biomarkers can predict a small proportion of HPV infections associated with carcinoma in situ. For example, an increased expression of the viral oncogenes *E6* and *E7*, which interfere with cell cycle control and apoptosis and induce chromosomal instability [11], is a hallmark of the transition from acute infection to carcinoma in situ. These oncoproteins also induce abnormal chromosome copy numbers and microRNA expression. Techniques for detecting HR–HPV *E6*/*E7* ribonucleic acid (RNA) have been developed; their sensitivity and specificity for detecting CIN 2 have been reported to be 71.4% and 75.8%, respectively [12,13,14,15,16]. Detection of *E6*/*E7* RNA may be more useful than HR–HPV DNA testing for diagnosing CIN 2+ and predicting disease progression [17]. Real–time multiplex nucleic acid sequence-based assays (e.g., the NucliSENS EasyQ HPV assay) show that HPV *E6*/*E7* RNA testing has a specificity of 50% and a positive predictive value (PPV) of 62% for CIN 2+, both of which are higher than the corresponding values for HPV DNA testing (specificity of 18% and PPV of 52%). The higher specificity and PPV of HPV *E6*/*E7* RNA testing are valuable in predicting insignificant HPV DNA infection among cases with borderline cytological findings [18]. Moreover, droplet digital PCR is more sensitive than real-time PCR for detecting HPV DNA and RNA [19,20]. Transcriptionally active HR-HPV in patients with head and neck squamous cell carcinoma (HNSCC) was previously visualized using a novel *E6*/*E7* RNA in situ hybridization (ISH) method [21,22]. Additionally, chromogenic ISH and p16^INK4A^/Ki–67 dual immunohistochemical staining on formalin-fixed paraffin-embedded (FFPE) cervical specimens correlated with *E6*/*E7* RNA expression [23]. Therefore, the detection of HPV *E6*/*E7* RNA, combined with human protein biomarker assays, may facilitate the diagnosis of abnormal cervical lesions and predict their progression.

In developed countries, molecular techniques for HPV DNA detection (e.g., Hybrid Capture^®^ 2) are combined with assays for host protein biomarkers, such as p16^INK4A^ and Ki–67, for the early detection of abnormal cervical lesions and the prediction of lesion grades; p16^INK4A^ is a surrogate biomarker for HPV in women with invasive cervical cancer, and its expression is highly associated with pathological grading [24,25]. The histologic evaluation of p16^INK4A^ and Ki–67 improves diagnostic accuracy [26]; dual staining was introduced mainly to increase the reproducibility and specificity of stand–alone p16^INK4A^ staining. Regardless of HPV status, diffuse p16^INK4A^ immunostaining is a hallmark of high-grade squamous intraepithelial lesions [27] and is an efficient screening tool [28]. Several candidate biomarkers and combinations thereof are being explored to predict the transition step [29]. However, which biomarkers should be used clinically remains unknown. At present, many clinicians and researchers continue to rely on traditional histological gradations—CIN 1, CIN 2, and CIN 3 (including carcinoma in situ); however, this approach is limited by subjectivity and poor reproducibility, especially in diagnosing CIN 1 and CIN 2 [30]. The accuracy of histopathological diagnosis is also limited by the tendency of colposcopic biopsies to miss small CIN 3 lesions almost 50% of the time [31]. Discovering biomarkers to clarify the risk of progression in the pathogenesis of cervical cancer is a major goal [32].

A preliminary study by our group found that the expression of HR-HPV *E6*/*E7* RNA is positively associated with the expression of cortactin. The *CTTN* gene, which encodes cortactin (the “cortical actin-binding protein”) is located on chromosome 11q13. Cortactin recruits Arp2/3 complex proteins and binds to actin microfilaments. It also promotes lamellipodia and invadopodia formation, cell migration, endocytosis, cell mortality, and tumor invasiveness [33,34]. Cortactin is overexpressed in many cancers [35,36] at high risk of invasiveness and metastasis, including hepatocellular carcinoma [37], colorectal cancer, glioblastoma, HNSCC, oral squamous cell carcinoma, lung squamous cell carcinoma, gliosarcoma, breast cancer, and melanoma [35]. Amplification of the *CTTN* gene and the resulting overexpression of cortactin have been observed in 15% of primary metastatic breast carcinomas and nearly 30% of HNSCCs [33,38]. Cortactin may also be associated with *E6*/*E7* RNA in HR-HPV-associated cervical cancer and is a potential diagnostic biomarker studied by our group. Furthermore, the majority of these highly sensitive techniques have not yet been introduced to clinical practice.

Over the last two decades, a variety of machine learning techniques and feature selection algorithms have been widely applied to determine disease prognosis and predict certain conditions [39]. These techniques are used in conjunction with logistic regression models to assess the importance of various genes. After important genes are identified, the same logistic regression model is then used for cancer classification and risk prediction [39]. Several prediction models are currently widely used in clinical practice, including the model for breast cancer incidence [40,41] and the predictive risk-scoring model for central lymph node metastasis [42]. The prediction model for breast cancer recurrence can be viewed at https://breast.predict.nhs.uk/predict.html (accessed on 8 March 2023). Mathematical models (MMs) are also used to determine the likelihood of relapse and predict responses to chemotherapy among patients with breast cancer [43], as well as to diagnose precancerous cervical lesions and predict progression [44,45].

In the comparison between histopathological method and modelling using multiple biomarkers, this study showed more advantages that can be used to derive scores for risk prediction, not only for the diagnosis of cervical lesion from similar biopsy samples. However, the current practice of histologic grading or using a single molecular biomarker can facilitate differential diagnosis.

Our study investigated the expression of cortactin in FFPE cervical specimens with diverse lesion grades in combination with other related biomarkers. Biomarkers p16^INK4A^ and Ki–67 were used as protein biomarker controls during immunohistochemical (IHC) staining. HPV *E6*/*E7* RNA ISH was also used. The relationship between IHC staining and ISH data was evaluated in association with clinical characteristics, and MMs were developed to estimate risk scores using linear regression analysis. Receiver operating characteristic (ROC) curves and areas under the curve (AUC) were used to identify the best MMs. Risk scores from the model were then used to predict the risk of abnormal or precancerous cervical lesion progression and may have utility in other cancer contexts in the future.

## 2. Materials and Methods

### 2.1. Specimens

Three hundred and sixty-three FFPE cervical tissue samples were collected from women who underwent routine cervical cancer testing with a colposcopy at Phayao Hospital, Phayao, Thailand in 2012 (233 samples) and 2013 (130 samples). This work was approved by the Human Research Ethics Committee of the University of Phayao (2/015/59) and Phayao Hospital (HE–59–02–0008). The sample size was calculated according to the known prevalence of HPV in the community as follows: N (case/age group) = Z^2^_1−a_ P(1 − P)/d^2^″. The required number of participants was calculated from a mean ± SD of 52.9 ± 32.1% of HPV prevalence, a Z of 1.96 for the 95% confidence level, and a d of 0.05 [46].

All FFPE cervical tissues were reviewed by two pathologists, and the following histopathological grades were assigned: normal (211 cases), low-grade squamous intraepithelial lesion (LSIL; 65 cases), HSIL (58 cases), and invasive cervical cancer (squamous cell carcinoma [SCC]; 29 cases). The 233 samples collected in 2012 were defined as the “test sample set” and the 130 samples collected in 2013 as the “confirmed sample set” (Table 1). The test sample set was used to develop the MM using a linear regression model, and the confirmed sample set was used to test the regression model.

### 2.2. Tissue Microarray (TMA) Preparation

The selected areas of the FFPE cervical tissues were stained with hematoxylin and eosin and graded by a pathologist according to the World Health Organization criteria. Paraffin tissue blocks were made by removing 1.5 mm cores of the tissues and organized into TMAs (Arraymold, Salt Lake City, UT, USA).

### 2.3. HR-HPV E6/E7 RNA Chromogenic ISH

*E6/E7* RNA chromogenic ISH was performed using the RNAscope 2.5 HD Detection Kit (BROWN) and Quick Guide for FFPE Tissues (Advanced Cell Diagnostics, Hayward, CA, USA) with specific combinations of *E6* or *E7* probes to detect 18 different HR-HPV types when low copy target gene expression was anticipated (1–20 copies per cell). The FFPE sections (5 µm) were de–paraffinized through xylene and ethanol washes and treated as follows: pre–treatment 1 (endogenous hydrogen peroxide block solution) for 10 min at RT; pre–treatment 2 for 45 min at 105 °C; and pre–treatment 3 (protease digestion) for 30 min at 40 °C. After the treatments, the sections were rinsed with water. The tissues were hybridized in a hybridization solution with *E6*/*E7* RNA chromogenic ISH probes in a moist chamber and without a cover slip for 2–3 h at 40 °C. Thereafter, the hybridized probe’s signal was amplified through the serial application of Amp 1 (pre–amplifier step), Amp 2 (signal enhancer step), Amp 3 (amplifier step), Amp 4 (label probe step), Amp 5, and Amp 6 (signal amplification steps); this was followed by the washing steps. Horseradish peroxidase (HRP) activity was then evaluated through the application of 3, 3′–diaminobenzidine (DAB) for 10 min at RT. The sections were then counterstained with hematoxylin, cleared in xylene, and mounted with Permount. The expression signal data were recorded according to negative and positive staining. The internal controls used for the RNAscope chromogenic ISH were proprietary probes for human sequence ubiquitin C (positive control to demonstrate detectable RNA in the FFPE samples) and *Bacillus subtilis* (*B. subtilis*) dapB RNA targets (negative control). Ubiquitin C staining was scored to confirm the presence of the signal and its intensity. *B. subtilis* dapB staining was reviewed to confirm the absence of staining.

### 2.4. IHC Staining

IHC staining was performed on the TMAs to determine the expression of cortactin, p16^INK4A^, and Ki–67. Briefly, following de–paraffinization and re–hydration, the tissue sections on the slides were antigen-retrieved using a target retrieval solution (citrate buffer, pH 6.0) at 105 °C in an autoclave for 30 min. Rabbit monoclonal anti–cortactin antibody clone Ep1922Y (Abcam, Cambridge, MA, USA), mouse monoclonal anti–human p16^INK4A^ clone D25 (EMD Millipore Corporation, Temecula, CA, USA), and Ki–67 monoclonal antibody clone 20Raj1 (eBioscience^TM^, Thermo Fisher Scientific, San Diego, CA, USA) were applied at dilutions of 1:200, 1:100, and 1:100, respectively, in 1× phosphate-buffered saline for 60 min. This was followed by incubation with secondary detection antibodies using the Genemed Power–Stain^TM^ 1.0 Poly HRP DAB Kit for Mouse + Rabbit (Sakura Finetek, Torrance, CA, USA). Immunostaining results were evaluated using light microscopy with a 40× objective, and both Allred score (AS; score and intensity of staining) and positive/negative status were recorded.

IHC staining patterns were scored in reference to the proportion of cells that stained: 0 = negative; 1 = rarely positive (<1%); 2 = focally positive (1–25%); 3 = variably positive (25–75%); and 4 = uniformly positive (>75%). In terms of the staining intensity, the IHC staining patterns were scored as follows: 0 = negative; 1 = weakly positive; 2 = moderately positive; and 3 = strongly positive. These scores were added to achieve an Allred scored (AS) ranging from 0 to 7 [47].

The criteria for distinguishing positive and negative IHC statuses are shown in Table 2.

### 2.5. Mathematical Models (MM) and Risk Score Development

A linear regression model in SPSS version 16 was used to develop the MMs for the expected cervical lesion grade. The model stepwise was Y = β0 + β1 X1 + β2 X2 + β3 X3 + … + βn Xn [48,49,50]. Y is the dependent variable, where 1 represents a normal value and 3, 4, and 5 represent LSIL, HSIL, and SCC, respectively. The independent variables (X, X_1_, X_2_, X_3_…X_n_) consisted of the five age groups (groups 1–5: 19–30, 31–40, 41–50, 51–60, and >60 years, respectively); IHC results of p16^INK4A^, Ki–67, and cortactin; and *E6*/*E7* RNA ISH results. The IHC positive/negative status was recorded as “2” (positive) or “1” (negative), whereas the AS of the staining intensity ranged from 0 to 7. The *E6*/*E7* RNA expression in the chromogenic ISH was recorded as “2” (positive) or “1” (negative) (Appendix A). B (β_0_, β_1_, β_2_, β_3_…β_n_) was the regression coefficient demonstrated as tolerance (0–1) and variance inflation factor (VIF; 1 to infinity). When the tolerance or VIF was near 1, a smaller association with the dependent variable (Y) was considered. When tolerance was near 0 or the VIF was high (>1), a high association with the dependent variable was considered. To develop the linear regression model, we used the test sample set (233 samples) (Table 1). The expected cervical lesion grade and risk score for the progression of abnormal cervical precancerous lesions were calculated.

### 2.6. Statistical Analysis

Statistical analysis was performed using SPSS version 16. The correlations between the cervical grade and the protein marker (positive/negative) were evaluated using the Pearson *Chi*-square test (significance level: *p* < 0.05). The correlations between the cervical grade and the protein marker (AS) were evaluated using one-way ANOVA (significance level: *p* < 0.05). The MM was included in the regression analysis (significance level: *p* < 0.05), and the ROC curves and AUC were evaluated using SPSS.

## 3. Results

### 3.1. Baseline Characteristics

A total of 363 FFPE cervical tissue samples were studied. These samples were retrieved in 2012 (233 samples) and 2013 (130 samples) from women aged 19–95 years. Table 1 shows the sample characteristics. The most common age group in both 2012 and 2013 was the 41–50–year age group. LSILs and HSILs were common based on the abnormal histopathological grades.

### 3.2. HR–HPV E6/E7 RNA Chromogenic ISH

Positive *E6*/*E7* RNA signals were mostly present in the cells (Figure 1). An increase in positive *E6*/*E7* RNA signals was associated with a severe grade of cervical lesions (Table 3). Table 4 shows the sensitivity, specificity, positive predictive value (PPV), and negative predictive value (NPV) of RNA *E6*/*E7* for detecting LSIL+ * and HSIL+ ** compared with normal cervical tissues. An association between p16^INK4A^ and RNA *E6/E7* was found in this study, which was consistent with Zappacosta R. et al. (2013) [32].

### 3.3. p16^INK4A^ and Ki–67 Immunostaining

The expression patterns of p16^INK4a^ and Ki–67 are shown in Figure 2 and Figure 3, respectively. A positive expression and the AS of p16^INK4a^ and Ki–67 were significantly associated with increasing severity grades of the cervical lesions, as shown in Table 3. Table 4 shows the sensitivity and specificity of p16^INK4A^ and Ki–67 for detecting LSIL+ and normal cases and for detecting HSIL+ and normal cases, respectively.

### 3.4. Cortactin Immunostaining

Five expression patterns of cortactin were observed according to their localization, staining, and intensity (Table 2). Positive cortactin staining was detected at significant levels in 84/211 normal cases (39.8%), 50/65 LSIL cases (76.9%), 46/58 HSIL cases (79.3%), and 24/29 SCC cases (82.8%). Figure 4 shows strong positive cytoplasmic overexpression staining (>75%, 2–3+) in some cases, with significant levels in 30/211 normal cases (14.21%), 28/65 LSIL cases (43.1%), 34/58 HSIL cases (58.6%), and 23/29 SCC cases (79.3%). Additionally, positive cortactin staining was characterized as a cytoplasmic membrane-positive staining pattern detected at significant levels in 10/211 normal cases (4.7%), 7/65 LSIL cases (10.8%), 3/58 HSIL cases (5.2%), and 10/29 SCC cases (34.5%). Meanwhile, the nuclear positive staining pattern was detected only in 1/58 (1.7%) HSIL case. The mean ASs were 2.7406, 4.4000, 5.0862, and 5.6552 for the normal cervical tissue, LSIL, HSIL, and SCC, respectively (Table 3). Positive cortactin staining was generally detected with significant differences across different grades of cervical lesions.

### 3.5. Mathematical Models (MM)

The MMs developed were simple linear regression models that included independent variables that consisted of age range (five categories) and the following biomarkers: cortactin, p16^INK4A^, Ki–67, and HPV *E6*/*E7* RNA. The coefficients of the independent variables in the best five MMs are shown in Table 5. Table 6 shows the five best MMs used for calculating the expected value (mean ± SD) for each cervical lesion grade shown in Table 7, Appendix A. The risk score for the prediction of abnormal cervical progression and precancerous lesion was calculated on the basis of the mean ± SD of each cervical lesion grade. Based on the expected values (mean ± SD), none of the models could differentiate between the normal and LSIL cases (*p* > 0.05); therefore, the normal cases were included with the LSIL cases in the confirmed sample set. In model 3, the risk score for the progression from LSIL was 2.60 (mean ± SD: 1.4843 ± 1.10780). The risk scores for the progression from HSIL (mean ± SD: 3.5374 ± 1.01427) and SCC (mean ± SD: 3.9516 ± 0.41838) ranged from 3.54 to 4.56 and from 3.95 to 4.37, respectively (Table 7). In model 4, the risk scores were 2.60 for LSIL, 3.62–4.85 for HSIL, and 4.23–4.76 for SCC. In model 5, the risk scores were 2.56 for LSIL, 3.56–4.66 for HSIL, and 4.03–4.48 for SCC. Appendix A shows the sensitivity and specificity of the five best models. Models 1–5 yielded a greater association with the variables with the disease outcome (OR) than did the other models. Models 3–5 revealed a great association with the variables with the disease outcome (OR). Models 2 and 3 could not differentiate between HSIL and SCC (*p* > 0.05). Interestingly, the predictive value of Models 1, 4, and 5 could significantly differentiate (1) the normal cases from HSIL–SCC (*p* < 0.001); (2) LSIL from HSIL–SCC (*p* < 0.001); and (3) HSIL from SCC (*p* < 0.05). The ROC curve and AUC of Models 3, 4, and 5 are shown in Appendix A. These MMs might have good value for the detection of cervical lesion progression and precancerous lesions.

Models 3, 4, and 5 were selected to assess the risk of abnormal cervical lesion progression and precancerous lesions in the confirmed sample set. The expected value (Y) was calculated in each case and compared with the risk score. When Y was equal to or lower than the risk score, the cervical lesions were suggested to have biomolecules characteristic of the baseline (i.e., normal tissues). When Y was higher than the risk score, the individuals were expected to be at risk of progression or to have “risk biomolecules.” Such individuals should be monitored. For example, when a histopathological LSIL case was evaluated by Model 3 and showed a predictive value of 1.59 (risk score: <2.60), the presence of LSIL with “baseline characteristic biomolecules” was suggested. However, when a histopathological LSIL case showed a predictive value of 3.02 (risk score: >2.60), it was suggested to be an LSIL case with present “risk biomolecules”. For the prediction of precancerous lesions in the normal and LSIL cases, the cases were predicted to have precancerous lesions when Y was higher than the risk score for HSIL (risk score: >3.54).

Appendix A demonstrate the prediction of cervical lesions using Models 3, 4, and 5. Model 4 showed the highest detection rate of cases with risk biomolecules in the LSIL (23/86 normal + LSIL cases, 26.7%) and HSIL groups (29/34 cases, 85.3%). The traditional histologic grading of the biopsies did not identify the 23 normal and LSIL cases. Without this knowledge, 23 patients would not have undergone close monitoring.

The next best models were Models 5 and 3. Model 4 best predicted the cases with precancerous lesions in the LSIL (5/86 cases, 5.8%) and HSIL groups (24/34 cases, 70.6%), while Models 3 and 5 predicted the cases with precancerous lesions in 3/86 (3.5%) cases in the LSIL group and 20/34 (58.8%) cases in the HSIL group. As shown in Table 8, the risk scores obtained by Models 3, 4, and 5 were suitable for detecting abnormal cervical lesions among patients in the LSIL group and for determining the risk of LSIL and HSIL. The ROC curve and AUC of Model 4 were significantly higher than those of Models 3 and 5 (*p* = 0.000) in terms of predicting the histopathological normal and LSIL cases with risk biomolecules and precancerous lesions (Appendix A). In the comparison between the sensitivity and specificity of Models 3 to 5 to distinguish between normal tissue and LSIL+HSIL in the confirmed sample set (Appendix A), the AUC values for predicting risk biomolecules were 0.757, 0.793, and 0.751, respectively. In the comparison between the sensitivity and specificity of Models 3 to 5 to distinguish between LSIL and HSIL in the confirmed sample set (Appendix A), the AUC values for predicting precancerous lesions were 0.777, 0.824, and 0.762, respectively.

## 4. Discussion

As previously reported, atypical cervical cells slowly grow and progress to precancerous lesions over a period of 10–20 years. Patients with these abnormal cells need to be monitored closely to prevent cervical cancer. In low–income countries, including Thailand, it is difficult to monitor these patients since they are usually lost in the follow–up. In this study, we were able to collect data from the initial presentations of our patients; however, we were unable to obtain follow–up results. Some of the patients might be at risk of developing cervical cancer.

Many studies have reported the clinical significance of p16^INK4A^ and Ki–67 expression as risk factors for cervical cancer. However, to date, no biomarkers have accurately predicted the progression of abnormal cervical cells and the development of precancerous lesions. The present study aimed to develop MMs and risk scores using a new biomarker, cortactin, combined with p16^INK4A^, Ki–67, and HPV mRNA. We intended to identify the best MM and risk score to predict the progression from normal cervical tissues to LSILs and HSILs and the risk of developing cervical precancerous lesions. We found that the sensitivity, specificity, PPV, and NPV of p16^INK4A^/Ki–67 were 68%, 69%, 61%, and 74% for detecting LSIL+ and 92%, 68%, 91%, and 96% for detecting HSIL+, respectively. These results are comparable to those of other studies. Li and colleagues found that the sensitivity, specificity, PPV, and NPV of p16^INK4A^/Ki–67 FFPE were 94%, 88%, 69%, and 98% for CIN 2+ detection, respectively, and 84%, 96%, 88%, and 96%, respectively, for CIN 3+ detection [51,52]. Among women with CIN 2, positive IHC staining for p16^INK4A^ and Ki–67 was strongly associated with disease progression [53].

Cortactin can promote cell migration, cell mortality, and tumor invasiveness in melanoma, colorectal cancer, and glioblastoma [33,34], and its expression was demonstrated to be significantly associated with poorer survival rates in patients with OSCC [54,55,56]. Meta-analyses have concluded that an overexpression of p16^INK4A^ [57,58] in cervical cancer relates to increased overall and disease-free survival rates, which differs from the function of cortactin. We found that cortactin staining (Table 3) might be a useful molecular diagnostic aid for cervical cancer screening, based on its sensitivity and specificity. However, the cellular functions of cortactin in cervical cancer require further investigation. The abnormal expression of cortactin was manifested both in intensity and localized distribution (Table 2). Correspondingly, a study of invasive and metastatic melanomas showed cortactin expression with a high density of (very strong) expression in SCC of 83% [59]. However, different distribution patterns of cortactin were also seen, such as in cases of nevi. This study reported that weak staining with low intensity was evenly distributed in the cytoplasm in normal nevi tissue and that strong staining was found in the cytoplasm of high-grade lesions. In contrast, strong staining was accentuated in the cell’s periphery in most melanomas. This was also seen in cultured melanoma cells, in which cortactin was distributed in the membrane ruffles and lamellipodia [59]. Therefore, the level of protein expression and the distribution of cortactin may reflect the abnormal upregulation of protein expression. The expression of cortactin in cervical cancer, which is reported for the first time by our group, may act as a biomarker for cervical cancer progression.

An increased expression of HR-HPV *E6* and *E7* correlates with the progression to high-grade lesions [60] and eventually to carcinoma in situ. These oncoproteins have been shown to induce abnormal chromosome copy numbers and miRNA expression in infectious processes [12,13,14,15,16]. The detection of HPV *E6*/*E7* RNA was combined with assays of biomarkers of human DNA, RNA, or protein for the diagnosis and prediction of abnormal cervical lesions. The sensitivity and specificity of HPV *E6*/*E7* RNA for detecting high-grade cytology (CIN 2) were 71.4% and 75.8% [12,13,14,15,16], respectively. The corresponding values for detecting CIN 2+ and CIN 3+ were 87.0% (75.6–93.6) and 88.0% (70.0–95.8), respectively. The specificity of HPV *E6/E7* RNA was 82.5% (77.3–86.8) for detecting CIN 2+ and 39.6% (34.0–45.5) for detecting CIN 3+ [59]. Herein, the sensitivity and specificity of HPV *E6*/*E7* RNA were 88% and 54% for predicting LSIL+, and they were 93% and 44% for predicting HSIL+, respectively (Table 4). The presence of HPV *E6*/*E7* RNA was associated with the future development of CIN 2+ among women with LSIL [60]. Moreover, the higher specificity (54% for LSIL+ and 44% for HSIL+) and NPV (81% for LSIL+ and 93% for HSIL+) of HPV *E6*/*E7* mRNA testing are valuable in predicting clinically insignificant HPV DNA infections and helping to avoid aggressive procedures (biopsies and over–referral for transient HPV infections), as well as for reducing patients’ anxieties and frequencies of follow up [18,61].

Several prediction models are currently widely used in clinical practice, including the model for breast cancer incidence, the Adjuvant Online Decision Aid [41,44,62] and that from http://www.predict.nhs.uk/predict.html (accessed on 8 March 2023), which uses MMs to determine the likelihood of relapse and to predict responses to chemotherapy for breast cancer [41]. Three of our five best MMs were evaluated using the confirmed sample set; Model 4. with risk scores of >2.60 and >3.62. showed the highest sensitivity for predicting risk biomolecules in the normal and LSIL cases and precancerous lesions, respectively.

The mean time for abnormal cervical cell progression from LSIL to HSIL with 8/45 (18%) oncogenic HPV types was 73.3 months (95% CI: 64.8–81.8 months). For non-oncogenic HPV (1/28, 4%), the mean time was 91.3 months (95% CI: 85.1–97.4 months), while for the 2/44 (5%) cases negative for HPV, the mean time was 83.5 months (95% CI: 78.0–89.1 months) [63]. In Model 4, 10/31 (32%) cases with LSIL and positive risk biomolecules included 25% of those with oncogenic HPV infection and 75% of those without HPV infection. Five patients with LSILs were younger than 25 years (3/5 cases, mean score >2.60). Bruno (2020) reported that the CIN 2 regression rates in women over 25 years of age are poor [64]. Herein, 26 patients with LSIL were older than 25 years (6/26 [23%]). Therefore, the risk score determined using Model 4 might predict the spontaneous regression or progression of LSIL [64] in women over 25 years of age. In addition, we found that 5/86 (5.8%) normal and LSIL cases with “risk biomolecules” were predicted to have precancerous lesions (>3.62), which might progress to cancerous lesions.

Our model also suggested that 13–14/34 (38–41%) cases of HSIL with “risk biomolecules” (3.95–4.23) might progress to cervical cancer. This is in broad agreement with the findings by Austin (2020), wherein they determined that only around 30% of CIN 3 lesions would progress to cervical cancer in 30 years [65]. However, this study found that slides suffer from issues such as the positions of the biopsies.

Wu et al. (2021) validated a prediction model in two cohorts in China with a follow-up duration of 3 years. In the first cohort, 42 cases were diagnosed as CIN 2+, with thirty-seven cases predicted to progress and five cases to not progress. In the second cohort, 28 cases were diagnosed as CIN 2+, with 11 cases predicted to progress and 17 cases to not progress [66]. Although this is a starting point for research using machine learning, our study demonstrates that machine–learning–based algorithms using input data from the expression levels of multiple biomarkers have potential for diagnosing and predicting disease progression [67,68] and consequently for solving health problems currently considered unsolvable, such as cancer.

## 5. Conclusions

MM-based analysis of the expression levels of multiple biomarkers, including p16^INK4A^, Ki–67, cortactin, and HPV *E6*/*E7* RNA, can provide a risk score for predicting the progression of abnormal cervical cells and the development of precancerous lesions in patients with normal histology and LSILs. For example, the relevant equation (Model 4) was Y = 0.535 + 0.387 (Ki–67^AS^) + 0.142(p16^INK4A AS^) + 0.530(cortactin^P/N^) + 0.506(RNA *E6*/*E7*^P/N^) − 0.786 (Ki–67^P/N^). These results suggest that monitoring patients with MM–based analyses of multiple biomarkers could help physicians design optimal therapeutic strategies and help predict cancer progression in the future.

## Figures and Tables

**Figure 1 diagnostics-13-01084-f001:**
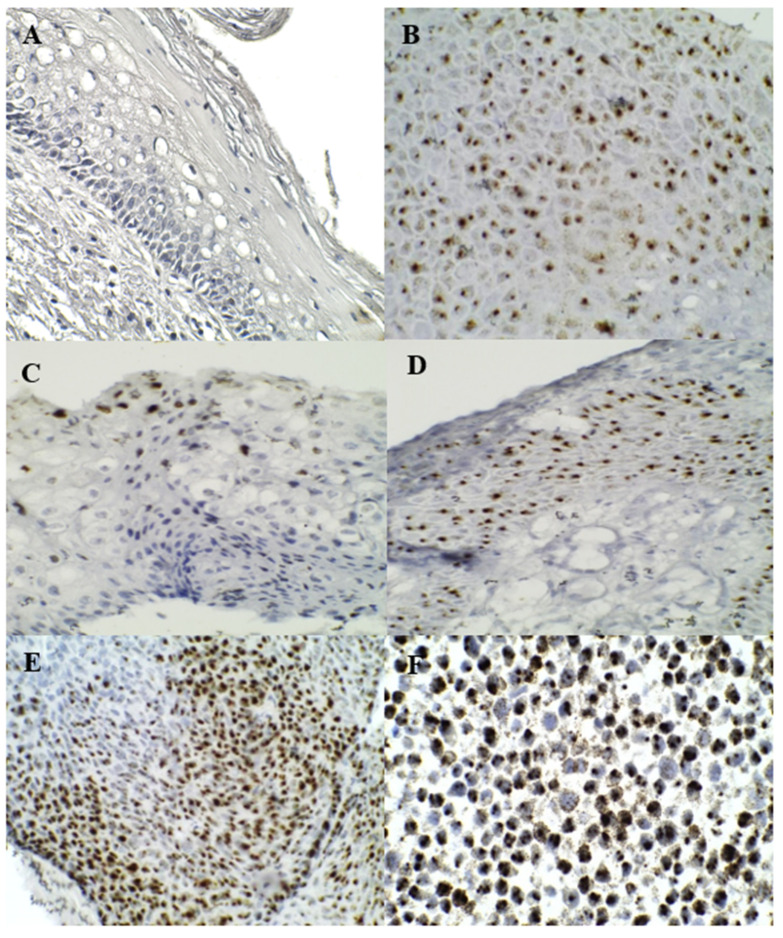
ISH staining of HPV *E6*/*E7* RNA: (**A**) was normal, (**B**,**C**) were LSIL, (**D**,**E**) were HSIL, (**F**) is positive cervical cell line control.

**Figure 2 diagnostics-13-01084-f002:**
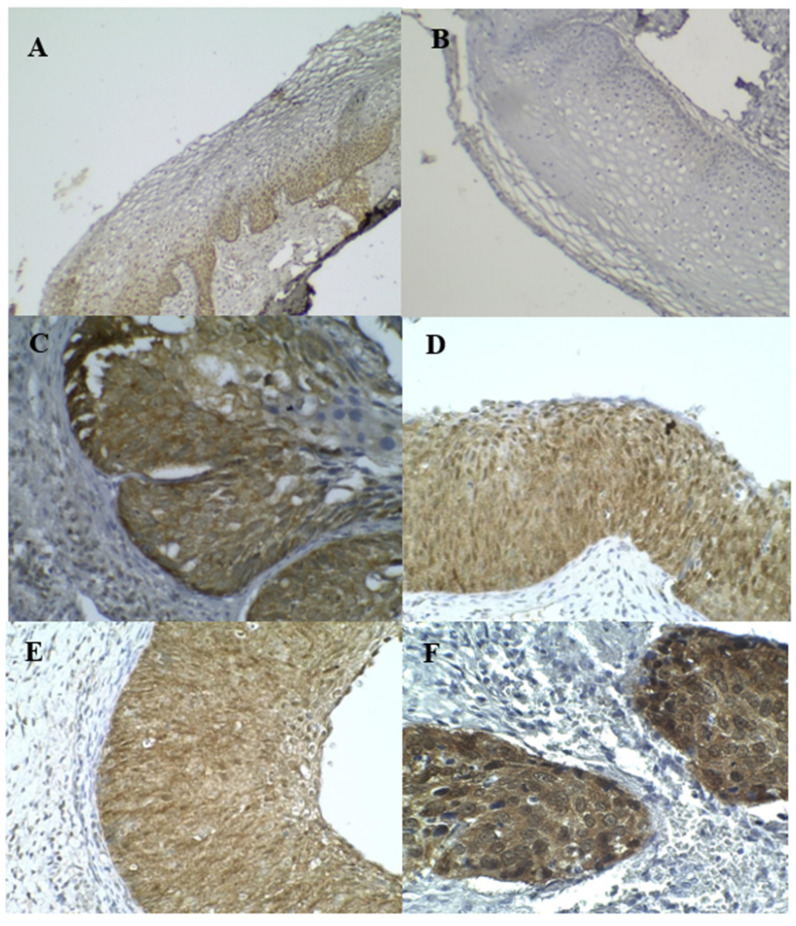
IHC staining of p16^INK4A^: (**A**,**B**) were normal, (**C**–**E**) were HSIL, (**F**) was SCC.

**Figure 3 diagnostics-13-01084-f003:**
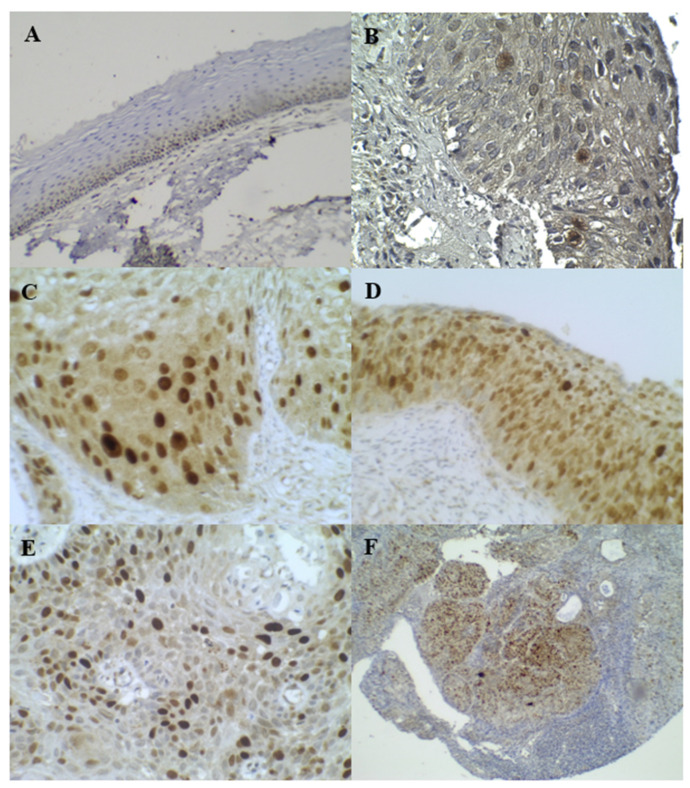
IHC staining of Ki–67: (**A**) was Normal, (**B**) was LSIL, (**C**,**D**) were HSIL, (**E**,**F**) were SCC.

**Figure 4 diagnostics-13-01084-f004:**
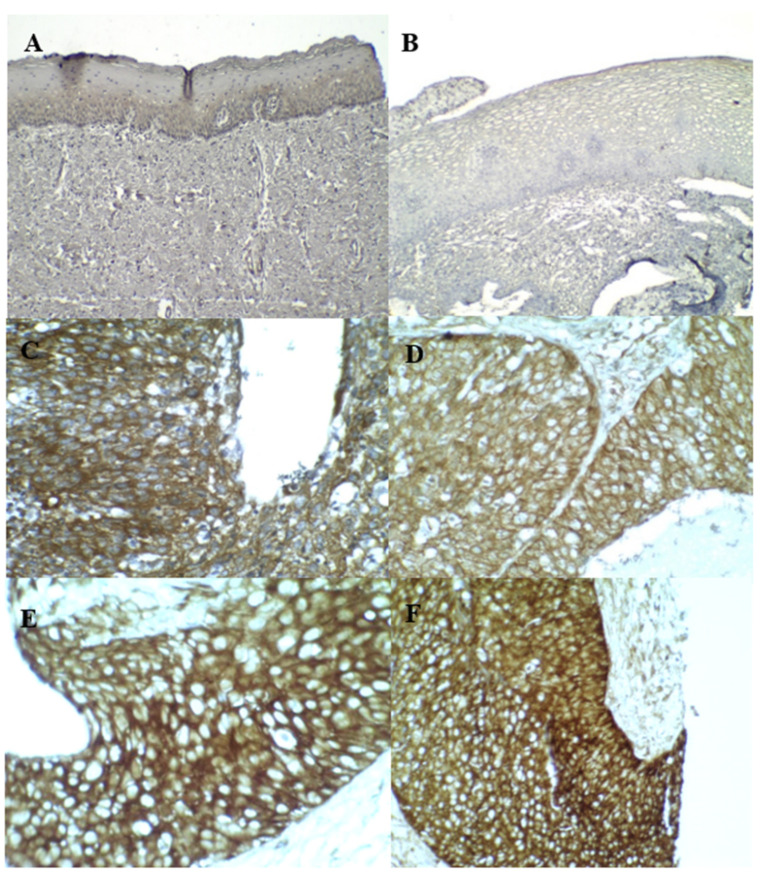
IHC staining for cortactin: (**A**) (weak cytoplasmic, normal cortactin expression) and (**B**) (negative) were normal. (**C**–**F**) were HSIL (positive cytoplasmic overexpression).

**Table 1 diagnostics-13-01084-t001:** Characteristics of the samples.

		Group 1 (233 Cases)	Group 2 (130 Cases)	Total (363 Cases)	*p*-Value
**Age (years)**	Mean	46.14	45.89	46.05	0.835 *
	SD	10.81	11.21	10.93	
**Age groups (years)**	19–30	11 (4.72%)	10 (7.69%)	21(5.78%)	
	31–40	56 (24.03%)	28 (21.53%)	84 (23.14%)	
	41–50	94 (40.34%)	43 (33.07%)	137 (37.74%)	
	51–60	54 (23.17%)	38 (29.23%)	92 (25.34%)	
	>60	19 (8.15%)	11(8.46%)	29 (7.98%)	
**Pathological grades**	Normal	156 (66.95%)	55 (42.30%)	211 (58.12%)	
	LSIL	34 (14.59%)	31 (23.84%)	65 (17.90%)	
	HSIL	24 (10.30%)	34 (26.15%)	58 (15.97%)	
	SCC	19 (8.15%)	10 (7.69%)	29 (7.98%)	

Note: * Student *t*-test, Group 1 = test sample set, Group 2 = confirmed sample set.

**Table 2 diagnostics-13-01084-t002:** Criteria for distinguishing positive and negative IHC statuses.

Biomarkers	Pattern of Expression	Interpretation
p16^INK4A^	(1) Staining was assessed as strong positive (block positive) according to the amount of uniform strong positive staining in the cytoplasm and nucleus in ~1/3 to 3/3 thickness, signal strength (which would appear as a dark brown color), and diffusion (the signal involved >50% of the epithelium).	“Positive”
	(2) Positive ambiguous results were further grouped into three patterns: (2.1) Strong/basal (strong, diffuse, continuous staining of the lower third of the epithelium without upward extension). (2.2) Weak/diffuse (weak, diffuse, discontinuous staining reaching at least two third of the epithelium). (2.3) Strong/focal (strong, focal, and discontinuous staining located at any level of the epithelium).	“Positive”
	(3) Negative results were defined as either the total absence of staining or weak, focal, and discontinuous staining.	“Negative”
Ki–67	Negative Ki–67 staining was defined as either the total absence of staining or weak basal staining.	“Negative”
Cortactin	(1) Negative, weak cytoplasmic and/or nuclear staining. (2) Weak focal staining in the cytoplasm or nucleus (heterogeneous).	“Negative” “Normal cortactin expression”
	(3) Uniformly strong cytoplasmic staining, “positive cytoplasmic overexpression,” or strong focal staining in the cytoplasm or nucleus (heterogeneous). (4) Uniform strong cytoplasmic staining, focal nuclear staining, and “positive nuclear and cytoplasmic overexpression”. (5) Strong cytoplasmic membrane staining.	“Positive”

Note: IHC = immunohistochemistry.

**Table 3 diagnostics-13-01084-t003:** Correlation between the pathological grades and IHC staining results (positive/negative status and mean ± SD Allred score).

Biomarkers		N	Normal (211 Cases)	LSIL (65 Cases)	HSIL (58 Cases)	SCC (29 Cases)	Total (363 Cases)	*p*–Value
**p16^INK4A^**	P/N	P	9 (4.2%)	7 (10.8%)	40 (69.0%)	28 (96.6%)	103 (28.4%)	0.000 *
		N	202 (95.2%)	58 (89.2%)	18 (31.0%)	1 (3.4%)	260 (71.6%)	
	AS	Mean	0.35	1.46	4.62	6.24	1.70	0.000 **
		SD	1.15	2.25	2.98	1.38	2.70	
**Ki-67**	P/N	P	52 (24.6%)	18 (27.7%)	50 (86.2%)	27 (93.1%)	147 (40.5%)	0.000 *
		N	159 (75.4%)	47 (72.3%)	8 (13.8%)	2 (6.9%)	216 (59.5%)	
	AS	Mean	1.1226	1.2923	5.2241	5.9655	2.1923	0.000 **
		SD	1.84	2.10	2.19	1.90	2.68	
**Cortactin**	P/N	P	84 (39.8%)	50 (76.9%)	46 (79.3%)	24 (82.8%)	204 (56.2%)	0.000 *
		N	127 (60.2%)	15 (23.1%)	12 (20.7%)	5 (17.2%)	159 (43.8%)	
	AS	Mean	2.74	4.40	5.09	5.65	3.64	0.000 **
		SD	2.36	2.45	2.27	2.68	2.62	
**RNA *E6*/*E7***		N	Normal (154 cases)	LSIL (62 cases)	HSIL (75 cases)	SCC (29 cases)	Total (320 cases)	*p*–Value
	P/N	P	71 (46.1%)	49 (79.0%)	69 (92.0%)	28 (96.6%)	217 (67.8%)	0.000 *
		N	83 (53.9%)	13 (21.0%)	6 (8.0%)	1 (3.4%)	103 (32.2%)	

Note: * Pearson *Chi*–Square, ** one–way ANOVA, P/N = positive or negative status, AS = Allred score, SD = standard deviation.

**Table 4 diagnostics-13-01084-t004:** Sensitivity, specificity, PPV, and NPV of several markers for detecting the pathological grades.

	Sensitivity	Specificity	PPV	NPV
**LSIL+ * vs. Normal**				
**p16^INK4A^**	49	96	89	72
**Ki-67**	63	75	65	74
**Cotactin**	79	60	59	80
**RNA *E6*/*E7***	88	54	67	81
**HSIL+ ** vs. Normal**				
**p16^INK4A^**	78	94	81	93
**Ki-67**	89	75	52	95
**Cotactin**	80	51	34	89
**RNA *E6*/*E7***	93	44	45	93

Note: PPV = Positive predictive value. NPV = Negative predictive value. LSIL+ * indicates cervical lesion grades of LSIL and more severe (HSIL and SCC). HSIL+ ** was cervical lesion grades as HSIL and SCC.

**Table 5 diagnostics-13-01084-t005:** Coefficients of the linear regression models (five best models).

	Model	Unstandardized Coefficients		Sig.	Collinearity Statistics	
		B	Std. Error		Tolerance	VIF
**Model 1**	(Constant)	1.150	0.101	0.000		
	p16^INK4A AS^	0.197	0.038	0.000	0.574	1.742
	Ki–67^AS^	0.269	0.039	0.000	0.574	1.742
**Model 2**	(Constant)	0.358	0.235	0.130		
	Ki–67^AS^	0.267	0.037	0.000	0.574	1.742
	p16^INK4A P/N^	0.172	0.037	0.000	0.555	1.801
	Cortactin^P/N^	0.570	0.154	0.000	0.938	1.066
**Model 3**	(Constant)	−0.346	0.314	0.272		
	Ki–67 ^AS^	0.245	0.037	0.000	0.556	1.799
	p16^INK4A AS^	0.152	0.036	0.000	0.539	1.854
	Cortactin^P/N^	0.557	0.150	0.000	0.937	1.067
	RNA *E6*/*E7* ^P/N^	0.518	0.159	0.001	0.841	1.189
**Model 4**	(Constant)	0.535	0.506	0.292		
	Ki–67^AS^	0.387	0.074	0.000	0.134	7.446
	p16^INK4A AS^	0.142	0.036	0.000	0.531	1.883
	Cortactin^P/N^	0.530	0.148	0.000	0.931	1.074
	RNA *E6*/*E7* ^P/N^	0.506	0.157	0.002	0.840	1.190
	Ki-67 ^P/N^	−0.786	0.356	0.029	0.166	6.040
**Model 5**	(Constant)	0.920	0.534	0.087		
	Ki–67^AS^	0.387	0.073	0.000	0.134	7.446
	p16^INK4A AS^	0.139	0.036	0.000	0.530	1.886
	Cortactin^P/N^	0.539	0.147	0.000	0.930	1.075
	RNA *E6*/*E7* ^P/N^	0.517	0.155	0.001	0.839	1.191
	Ki–67^P/N^	−0.747	0.353	0.036	0.165	6.057
	Age groups	−0.153	0.073	0.039	0.987	1.014

Note: P/N = positive or negative status, AS = Allred scored, B = regression coefficient, Std. Error = Standard Error, VIF = variance inflation factor.

**Table 6 diagnostics-13-01084-t006:** Equations of the linear regression models for calculating the expected value.

Model	Equations
Model 1	Y = 1.150 + 0.197 (p16^INK4A AS^) + 0.269 (Ki–67^AS^)
Model 2	Y = −0.358+ 0.267 (Ki–67^AS^) + 0.172 (p16^INK4A P/N^) + 0.570 (Cortactin^P/N^)
Model 3	Y = −0.346 + 0.245 (Ki–67^AS^) + 0.152 (p16^INK4AAS^) + 0.557 (Cortactin^P/N^) + 0.518 (RNA *E6*/*E7* ^P/N^)
Model 4	Y = 0.535 + 0.387 (Ki–67^AS^) + 0.142 (p16^INK4A AS^) + 0.530 (Cortactin^P/N^) + 0.506 (RNA *E6*/*E7* ^P/N^) − 0.786 (Ki–67 ^P/N^)
Model 5	Y = 0.920 + 0.387 (Ki–67 ^AS^) + 0.139 (p16^INK4A AS^) + 0.539 (Cortactin ^P/N^) + 0.517 (RNA *E6*/*E7* ^P/N^) − 0.747 (Ki–67 ^P/N^) − 0.153 (Age groups)

Note: P/N = positive or negative status, AS = Allred scored. Means of the expected values from five linear regression models in clinical pathological grades (each sample) of confirmed sample sets were shown in Appendix A.

**Table 7 diagnostics-13-01084-t007:** Means and SDs of the expected values from the linear regression models and clinical pathological grades of the test and confirmed sample sets.

	Test Sample Set					Confirmed Sample Set			
		N	Mean	SD	*p*–Value	N	Mean	SD	*p*–Value
**Model 1**	Normal	156	1.4913	0.53391	0.000	55	1.6146	0.5864	0.000
	LSIL	34	1.6402	0.87461		31	1.9450	0.8526	
	HSIL	24	3.4301	1.01324		34	3.4906	1.0748	
	SCC	19	3.9961	0.39674		10	3.9619	0.7745	
	Total	233	1.9170	1.06639		130	2.3646	1.2106	
**Model 2**	Normal	156	0.8589	0.49971	0.000	55	1.0939	0.7041	0.000
	LSIL	34	1.0008	0.63920		31	1.3892	0.7130	
	HSIL	24	2.3430	0.73615		34	2.3541	0.7170	
	SCC	19	2.6392	0.33303		10	2.5678	0.7772	
	Total	233	1.1770	0.82221		130	1.6073	0.9170	
**Model 3**	Normal	156	1.2044	0.65480	0.000	55	1.2069	0.8686	0.000
	LSIL	34	1.4843	1.10780		31	2.1960	0.9162	
	HSIL	24	3.5374	1.01427		34	3.3083	1.1946	
	SCC	19	3.9516	0.41838		10	3.9686	0.7087	
	Total	233	1.7096	1.22683		130	2.2048	1.3778	
**Model 4**	Normal	156	1.4316	0.73041	0.000	55	1.7865	0.9271	0.000
	LSIL	34	1.6542	0.98381		31	2.0258	1.1043	
	HSIL	24	3.6226	1.23028		34	3.8366	1.1726	
	SCC	19	4.2301	0.52538		10	4.2166	1.2110	
	Total	233	1.9180	1.25429		130	2.5667	1.4364	
**Model 5**	Normal	156	1.2014	0.62799	0.000	55	1.1862	0.8520	0.000
	LSIL	34	1.4728	1.08656		31	2.2305	0.9044	
	HSIL	24	3.5554	1.10289		34	3.3733	1.2589	
	SCC	19	4.0282	0.45701		10	3.9481	0.7920	
	Total	233	1.7140	1.24210		130	2.2197	1.4076	

Note: one–way ANOVA.

**Table 8 diagnostics-13-01084-t008:** Sensitivity, specificity, PPV, and NPV of the pathological grade in Models 3 to 5 in the confirmed sample set.

	Risk Biomolecules Prediction					Precancerous Lesion Prediction				
	LSIL Group (Normal + LSIL) vs. HSIL					LSIL Group (Normal + LSIL) vs. HSIL				
	Sensitivity	Specificity	PPV	NPV	OR	Sensitivity	Specificity	PPV	NPV	OR
**Model 3**	68	84	62	87	10.8	59	97	87	86	39.5
**Model 4**	85	73	56	93	15.9	71	94	83	89	38.8
**Model 5**	68	83	61	87	9.9	56	97	86	84	35.0

Note: PPV = Positive predictive value, NPV = Negative predictive value, OR = odds ratio.

## Data Availability

Not applicable.

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
