# Peer review of "Mathematical Modelling of Cervical Precancerous Lesion Grade Risk Scores: Linear Regression Analysis of Cellular Protein Biomarkers and Human Papillomavirus E6/E7 RNA Staining Patterns"

_diagnostics, 2023, doi:10.3390/diagnostics13061084_

Round 1
Reviewer 1 Report
The manuscript submitted by Bumrungthai and co-authors describe a methodology to assess biomarkers of cervical lessions with a mathematical methodology. The manuscript is fairly clear but it misses the most important point, that is to assess how good or bad this proposal is, we need a comparison. The paper several times mentions that the problem of histological grading may be reproducibility or problems obtaining the biopsy, but it is not clear how much better can the modelling be especially if the biomarkers may come from biopsies themselves.
Specific comments
A paper that focuses on coploscopic histology should have included at least one figure about colposcopic histology! This is surprising to have to be even mentioned. Given that the manuscript motivation states incorrect biopsy placement or the reproducibility problems of traditional histological grading, these two should have been illustrated so that the reader (especially in a journal called Diagnostics) can understand the importance of what will be later proposed. An example of an incorrect biopsy placement, grades 1,2,3 and cases where there are errors should be presented. I happen to be familiar with Ki-67 and H&E, but perhaps not every reader, and I also fail to see exactly which cases (of images) could be improved with these biomarkers. What really leaves me more puzzled is that it seems that the biomarkers are extracted from the imaging itself (“The IHC value was recorded as “2” (positive) or “1” (negative), whereas the AS of 210 the staining intensity ranged from 0 to 7.”) which then makes me wonder if these biomarkers come from images that could be used for grading, and these images suffer from problems like the position of the biopsy, then what advantage they present?
The numbers presented in the abstract do not make much sense, 23/86 have molecules that suggest risk of progression, but what is really important is to know how much better or worse is the methodology proposed over traditional histological grading or biopsies, which were previously cited as problematic. Without this knowledge, the conclusion (“These results suggest that risk score based on the biomarker levels analyzed 35
might predict the risk of cervical lesion progression and precancerous lesion development”) is really meaningless.
LSIL is mentioned in the abstract and introduction without defining the acronym, same for RNA, this last one may be really familiar but it is best to always define acronyms first time they appear
Model 4 is mentioned without any reference to what it is or what the other 3 (or more) are.
The phrase “Combining p16INK4A 80 and Ki-67 with histology improves the diagnostic accuracy” is not correct, it should be corrected, you do not combine p16 and Ki-67, you also do not combine with histology.
“Regardless of the HPV status, diffuse p16INK4A immunostaining is 83 the hallmark of high-grade squamous intraepithelial lesions (HSILs) [21] and might be an 84 efficient screening tool [22]” Might be? If it is not, why not? Why not illustrating the document with some immunostaining examples so that the advantage of the authors proposal can be better appreciated.
Reviewer 2 Report
Journal – Diagnostics
Manuscript diagnostics-2148361
Type research article
Title. Mathematical modelling of cervical precancerous lesion grade risk scores: linear regression analysis of cellular protein biomarkers and human papillomavirus E6/E7 RNA staining pattern
Main comment.
This work aimed to develop mathematical models and risk scores for predicting the risk of cervical lesion progression and precancerous lesions among patients in northern Thailand by evaluating the expression of numerous biomarkers. As a main comment I suggest reducing the length of the introduction for an easier reading.
Additional comments:
§ Authors are kindly encouraged to include the following references in the introduction:
1. https://doi.org/10.1002/jcp.24808
2. https://www.ncbi.nlm.nih.gov/books/NBK544371/
3. PMID: 31335091
4. PMID: 35008799
5. PMID: 24901204
§ Are the authors referring to CIN1? (line 47)
§ Droplet digital PCR methods, which are more sensitive than real time methods), have also been developed for detecting HPV DNA/RNA molecules (lines 64-69)
§ In the introduction, the authors should remark the fact that the majority of these highly sensitive techniques have still not introduced in the clinical practice
§ Please check this typo mistake in table 1 “HSIL 24 (1030%)”
§ Methodologies, including the linear regression model, are lacking in supporting references
§ Has the correlation between p16INK4A and RNA E6/E7 been evaluated? p16INK4A is a well known surrogate marker of HPV infection https://www.mdpi.com/2076-393X/10/2/204
§ Please avoid abbreviations in subhead titles (line 282)
Round 2
Reviewer 1 Report
The authors have addressed my comments and improved the quality of the manuscript